# Pancreatic Cancer Organoids: An Emerging Platform for Precision Medicine?

**DOI:** 10.3390/biomedicines11030890

**Published:** 2023-03-14

**Authors:** Evangelia Sereti, Irida Papapostolou, Konstantinos Dimas

**Affiliations:** 1Department of Translational Medicine, Lund University, 22363 Lund, Sweden; 2Department of Biochemistry and Molecular Medicine, 3012 Bern, Switzerland; 3Department of Pharmacology, University of Thessaly, Biopolis, 41500 Larissa, Greece

**Keywords:** pancreatic organoids, pancreatic cancer, precision medicine

## Abstract

Despite recent therapeutic advances, pancreatic ductal adenocarcinoma (PDAC) remains one of the most aggressive malignancies, with remarkable resistance to treatment, poor prognosis, and poor clinical outcome. More efficient therapeutic approaches are urgently needed to improve patients’ survival. Recently, the development of organoid culture systems has gained substantial attention as an emerging preclinical research model. PDAC organoids have been developed to study pancreatic cancer biology, progression, and treatment response, filling the translational gap between *in vitro* and *in vivo* models. Here, we review the rapidly evolving field of PDAC organoids and their potential as powerful preclinical tools that could pave the way towards precision medicine for pancreatic cancer.

## 1. Introduction

Despite significant progress regarding our knowledge of cancer biology, cancer remains a leading cause of death worldwide. In the European Union (EU-27), according to recent data (22 July 2020) released from the European Cancer Information System (ECIS), cancer burden has risen to 2.7 million new cases (all types, excluding non-melanoma skin cancer) with 1.3 million deaths in 2020 [1]. The above also records that pancreatic cancer (PC) is the seventh most common cancer for both sexes, with an incidence of approximately 3.5%, and the fourth most common cause of cancer related death for both sexes and all ages (with an estimated percentage of 7.1%) while it rises to the third place among individuals over the age of 65. Pancreatic ductal adenocarcinoma (PDAC) represents the 85% of all cases of pancreatic cancer [2]. Smoking, even after cessation, diabetes mellitus, obesity, age, and genetic predisposition are the main risk factors identified so far to be implicated in the development of PDAC [3,4,5,6]. Despite constant efforts to develop novel diagnostic tools and treatment approaches, it is estimated that PDAC will rise to the second most common cause of cancer death by 2030 [6,7]. The prognosis for PDAC patients remains unfavorable, with a 5-year overall survival rate of less than 5% [8]. Surgical resection followed by adjuvant therapy is the only therapeutic option; however, less than 15% of patients present local and potentially operable disease, while it is well known that operability and survival are better in patients with smaller lesions [9]. Patients with advanced disease at presentation or recurrence may receive palliative chemotherapy, although the rates of response and overall benefit are modest. The aggressive nature of PDAC and primary or secondary drug resistance [10] contribute significantly to this dismal prognosis. It has become evident that PDAC has a genetic component, with several relevant mutations, while its progression is characterized by high heterogeneity [11,12,13]. KRAS, TP53, CDKN2A, and SMAD4 have been identified as recurrently mutated genes in pancreatic cancer [14]. Pancreatic cancer progression and resistance are complex processes involving multiple mechanisms, including genetic and epigenetic alterations, tumor microenvironment, metabolic reprogramming, immune evasion, and tumor heterogeneity. Recent studies have shown that several protein kinases, like cyclin-dependent kinase 1 (CDK1) [15,16,17,18], play a significant role in pancreatic cancer resistance by modifying proteins involved in key signaling pathways. These modifications can lead to the activation of cellular processes that promote cancer cell growth, survival, and resistance to chemotherapy.

Due to the reasons stated above, the scientific community is focusing on prevention, by identifying and minimizing environmental risk factors, and of course on early diagnosis, which holds the promise of improved outcomes. Chemotherapy remains the cornerstone for PC management along with tumor resection, radiotherapy, and more recently immunotherapy. However, chemotherapy is severely restricted by multiple constraints, such as severe side effects and toxicity leading to non-compliance with prolonged treatment, drug resistance, incomplete cure, and low patient quality of life. Thus, there is an urgent need to develop novel and more effective therapies capable of improving PC patients’ survival rates.

To overcome these constraints and improve the overall outcome of chemotherapy, one of the major advances in cancer treatment focuses on the application of more personalized approaches. Personalized or precision medicine is a rapidly evolving approach that allows clinicians to select treatments that carry the potential to greatly improve both treatment effectiveness as well as patients’ quality of life. Precision medicine (which is the preferred term nowadays, for more details over the debate see [19]) is the result of many years of research on tumor biology that has led scientists to understand that individual differences in genetics, lifestyle, and environment will influence the way cancer patients respond to treatment.

## 2. Current Status

Beyond any doubt, the most successful example of precision medicine is pharmacogenomics. Pharmacogenomics is “the study of how genomic variation within the individual or their disease (including gene expression, epigenetics, germline, and somatic mutations) influences his/her response to drugs”, as described in the Precision Medicine Glossary of ESMO [19]. In pharmacogenomics, as explained in the Glossary, genomic variation is correlated with pharmacodynamics and pharmacokinetics. Pharmacogenomics aims to optimize drug therapy by maximizing therapeutic effects and minimizing adverse effects. However, these therapies are very often limited to narrow subsets of patients with a specific, molecularly defined tumor profile [20].

The efforts to develop chemosensitivity tests that could predict the response of tumors to anticancer drugs are not new. They date back to the mid-50s when various techniques for testing chemosensitivity were developed using tumor cells from cancer patients in *in vitro* experiments [21]. Human-derived cell lines are the most widely used models for these experiments. Such experiments are described in the paper by Keepers et al. [16], where the authors compare two assays commonly used for drug screening, the tetrazolium (MTT) and the sulforhodamine B (SRB) assays, in two established cancer cell lines: the HT29, a colorectal cancer cell line and the 11B, a squamous cell carcinoma cell line of the head and neck. The first human pancreatic cancer cell line was generated in 1963 [22], and since then many human PDAC cell lines have been established. Cell lines possess critical advantages, as they are easy to manipulate, they can grow indefinitely at low cost, and are suitable for high throughput pharmacological screening and genetic testing. However, cell line models also have significant drawbacks, the most important of which is their inability to recapitulate the histological and biological complexity of a tumor. Especially the absence of the tumor microenvironment, mainly consisting of extracellular matrix (ECM) components and various other cell types, such as fibroblasts, nerves, immune cells, adipocytes, and endothelial cells, is a critical limitation of the use of cell lines [23].

Nonetheless, efforts have been made towards the development of chemotherapy sensitivity and resistance assays with the use of patient-derived tumor cells that could be part of routine oncology practice [24]. These assays differ from the older approaches that quantify drug sensitivity by growth inhibition of tumor cells (such as tumor clonogenic and [^3^H]thymidine incorporation assays) to the newer approaches, i.e., the differential staining assay and the ATP bioluminescence assay (ATP-TCA), that measure the rate of tumor cell death as a readout [25]. These latter assays use the quantification of cell viability as a measure of anticancer drug effectiveness, whereas growth inhibition assays often promote the development of single clones, thus failing to reflect the *in vivo* situation [26]. ATP-TCA has been shown to comprise high sensitivity and reproducibility [27]. Prospective studies using this approach revealed a correlation between *in vitro* sensitivity and *in vivo* tumor response in metastasized breast cancer [28] and recurrent ovarian cancer [29]. However, the American Society of Clinical Oncology Working Group on Chemotherapy Sensitivity and Resistance Assays stated that chemosensitivity assays should not be recommended for clinical use since the current evidence is not enough to justify the use of any of these assays as a guide to personalized treatment of cancer patients [24].

Animal cancer models derived from human cancer cell-developed models (xenografts) are also widely used in translational research. However, these animal models have several serious pitfalls (limited potential to mirror the actual tumor microenvironment, intratumoral clonal heterogeneity and human stromal cells properties), parameters that greatly affect tumor invasion, migration, recurrence, and drug resistance. As an alternative, Patient-Derived Xenografts (PDX) developed by the engraftment of patients’ excised tumors directly into immune-deficient animal models, are novel preclinical tools that could achieve a greater similarity to human cancer genetics, tumor heterogeneity, and microenvironment. However, so far, only a limited number of studies have assessed PDX models as preclinical models for targeted chemotherapy and precision medicine. These studies have documented that even these sophisticated animal models entail serious disadvantages, such as the cost, tumor take-rate limitations and slow growth, that are important drawbacks for real-time personalized medicine approach [30].

A methodology that aims to bridge the utilization of human cancer cells cultures (or 2D cultures) and xenografts, which gained increasing popularity in basic and translational cancer research lately, is the development of three-dimensional cultures (3D cultures). These cultures aim to rebuild the 3D microenvironment of a tumor to cultivate an “identical twin” of a tumor in a culture dish that could be used for basic and translational research purposes, such as the anticancer drug testing. The first approach of this method is the cultivation of formations called “spheroids”. Historically, the first spheroid model was reported by Sutherland in 1986 [31]. Sutherland developed this model to study several microenvironmental factors associated with tumor growth. Since then, spheroids have been grown from a variety of normal cells, established tumor cell lines, or patient-derived tumor cells using different culture techniques and various artificial and natural extracellular matrices (ECM) and mechanical methods to generate spherical cell clusters [32,33,34,35]. Several studies have tested the usefulness of multicellular spheroids in cancer biology and the development of anticancer drugs. Importantly, these studies documented significant differences between 2D and 3D multicellular spheroid cultures in cancer drug sensitivity [36]. Spheroids have also been used for screening putative new anticancer compounds and for testing individual chemosensitivity in the context of cancer precision chemotherapy [37]. It remains to be comprehensively demonstrated that chemosensitivity data derived from 3D cell cultures capture clinically relevant responses. Moreover, the major limitation of these systems is that they cannot completely mimic the complex tissue architecture and the high degree of variability of individual tumors. Having noted that, 3D systems undoubtedly increase the number of cell-to-cell interactions and more closely resemble the architectural organization of tissues *in vivo*. Hence, the data gathered so far support the use of spheroids as reliable models of *in vivo* solid tumors and as drug screening platforms.

As spheroids have indeed proven to be useful in the development of anticancer drugs, researchers put effort towards improving this model. One of the most recent advances in this field is the development and introduction of more advanced forms of spheroids, which are called organoids.

## 3. Organoids

The main strategy to develop 3D cultures is to avoid cell attachment to the bottom of the culture dishes. This can be achieved simply by either keeping cells in suspension or by culturing them in special matrices. 3D cultures derived this way from monolayer cell cultures, are referred to as spheroids and share some characteristics with tumors *in vivo* (e.g., the production of ECM, increased chemo-resistance, the appearance of polarized cell junctions, etc.). However, the use of monolayers in this 3D cell culture system is reported to have limited *in vivo* relevance [38]. In a breakthrough study in 2009, Sato et al. [39] described the establishment of long-term culture conditions under which organoids could be developed and cultured by stem cells. They utilized a culture system in which specific niche factors were artificially supplied, including the Wnt agonist R-spondin 1, epidermal growth factor (EGF), and Noggin (a protein that is involved in the development of many body tissues, such as nerve tissue, muscles, and bones). Τo prevent cells undergoing anoikis, they used laminin rich Matrigel to support intestinal epithelial growth. This was the first successful attempt to grow organ-like structures for an extended period, up to 14 months. A month later, Ootani et al. [40] described the development of a robust long-term methodology for culturing small and large intestines of mouse origin (C57BL/6 mice), incorporating an air-liquid interface (ALI) and underlying stromal elements. In their method, Ootani and colleagues used a type I collagen gel as the scaffold to support the growth of intestinal cells from minced tissues, along with residual stromal components from tissue fragments that provided the essential niche factors. In 2014, two groups, Li et al. and Gao et al., described breakthrough advances in the use of organoid models in cancer research. Li et al. [41] used a single ALI culture method without modification to engineer oncogenic mutations into primary epithelial and mesenchymal organoids from the mouse colon, stomach, and pancreas. Their studies demonstrated the general utility of a primary organoid system (of mice origin though) for cancer modelling and driver oncogene validation in diverse gastrointestinal tissues. Gao et al. [42] used a 3D organoid system that allowed them the successful long-term culture of prostate cancer organoids from biopsy specimens and circulating tumor cells. Interestingly, when these organoids were grafted into SCID mice, the tumors recapitulated both the histological and immuno-histological patterns of the patient samples. The authors also performed growth assays to determine the sensitivity of organoids to enzalutamide and two PI3-kinase pathway inhibitors, everolimus, and BKM-120. This was practically the first attempt to incorporate organoids into the anticancer drug development research and precision medicine frame. The most important pitfall of this study was the low initial success rate, with only 6 out of 32 attempts from seven patients resulting in successful organoid development, yielding a success rate of only 15–20%.

Based on the findings of these pioneer studies, an “organoid” refers to a group of cells growing in a 3D structure that is generated directly from primary tissues (healthy or diseased, like in tumors), embryonic stem cells, circulating tumor cells, or pluripotent stem cells. Organoids are capable of self-renewal and self-organization, enabling them to recapitulate the histology and function of the tissue of origin. Nowadays, organoids can be maintained through an indefinite passage, stored in liquid nitrogen for future use, and, most importantly, preserve genetic stability [43]. 3D organoids share many characteristics with spheroids (3D spheres); however, the two systems are quite different. Spheroids originate from monolayers of epithelial cells (usually from established cell lines and rarely from patients’ tumor cells) with no specific supplements and with the major requirement to be cultured under low-adhesion culture conditions that promote cell self-aggregation. On the contrary, 3D organoids come directly from tissues (normal or diseased), adult stem cells, or embryonic stem cells, following specific culture protocols (Table 1). They retain self-renewal/self-organization capacities and can very closely recapitulate the architecture and characteristics of the tissue of origin. Organoids derived from clinical tumor specimens can also be passaged over time and maintain some of the molecular characteristics of the original tumor and are named tumoroids or tumor organoids [19].

### Methods of PDAC Organoids Development

A summary of the procedures followed by researchers to create pancreatic organoids is presented in Table 1. For an organoid to be created, samples from patients with PDAC are required. The samples are biopsies collected after the surgical removal of the tumor. 

There is a specific procedure for manipulating the specimen collected from the patient to successfully create PDAC organoids. Currently, there are two main approaches for developing organoids (Figure 1). In one approach, developed and established by Boj S et al., researchers cultivated organoids not directly from patients but by using PDX as an intermediate step [44,48]. According to this approach, after the tumor was surgically removed, it was transferred into a 50 mL tube containing RPMI 1640 (Roswell Park Memorial Institute) supplemented with 1% penicillin/streptomycin and 1% fungizone DCU. The sample was subsequently cut into small pieces to create fragments with a size < 2 mm. It should be noted that all procedures must take place within 24 h to maintain maximum viability of tumor cells. The next step was the *in vivo* development of the PDX. Specific strains of mice were chosen, such as the NSG mice, for the tumor to grow. The injection of the tumor fragments was performed either orthotopically (meaning that the fragments are directly injected into the organ of interest) or fragments were subcutaneously implanted. Hereafter, the mice were observed for the development of tumors. The tumors were measured regularly until they reached the desired, non-harming for mice, size. At the end of the procedure, mice are humanely euthanized, and the tumors are removed. After the extraction, the tumors were washed with special buffers for contaminating particles to be removed. Next, the tumors were sliced into small pieces with sterile forceps and blades. The pieces were digested with a special buffer for around 15 min at room temperature (RT). After digestion, the tumor pieces were passed through a strainer, and the process was repeated 2–3 times with washing buffers. These steps are needed to isolate as many cells as possible. The tube containing the desired cells was centrifuged at 1000 RPM for 5 min at 4 °C. After the centrifugation, the medium was carefully removed to avoid pellet disruption. The procedure followed from now on is similar for both techniques used. One difference is the consistency of the medium used for the organoid culture. The difference lies in the presence or absence of specific factors in the medium (Table 2). Some of these factors are the following: (1) EGFR, (2) L-glutamine, (3) amphotericin b, (4) Insulin transferrin selenium, (5) PDGF, and (6) IGF-1. Organoid cultures generally require a change of medium once per week and can be expanded by passaging at a 1:3 to 1:6 ratio once per week.

## 4. Organoids and Tumoroids in the Battle against Pancreatic Cancer 

The first to describe the system of organoids for pancreatic tissue was once again the Clever’s lab. In 2013, Huch et al. published their pioneer work, where they developed pancreatic organoids by isolating pancreatic ducts from the bulk of the pancreas of mice [49]. Initially they mixed, seeded, and cultured these pieces with Matrigel, using a culture medium based on AdDMEM/F12 and supplemented with B27, N-Acetylcysteine, gastrin, and the growth factors EGF, RSPO1-conditioned media, Noggin, FGF10, and Nicotinamide. They were able to expand and passage these organoids for at least 9 months and store them in cell culture freezing medium. Interestingly, the organoids were ready to be passaged within one week after seeding and culturing primary isolated ducts in Matrigel. Two years later, Boj et al. reported the first tumor organoids from human PDAC. To achieve this, the team substantially modified the procedure and culture conditions compared to mouse-derived organoids (Table 1) [44]. The material they used to develop human PDAC organoids was collected either from surgically excised tumors or by FNA (fine needle aspiration) biopsies, which is important considering that the vast majority of PDAC patients are inoperable [50]. The success rate of organoid formation was up to 85%, and they could be passaged indefinitely and cryopreserved for future use. Even more, these organoids showed complex morphology with differing degrees of dysplastic tall columnar cells resembling low-grade pancreatic intraepithelial neoplasias (PanINs). Further thorough characterization of these organoids suggested that they were indeed very closely mirroring neoplastic human pancreatic ductal cells, hence offering a model system to explore human pancreatic cancer biology. The orthotopic transplantation into *Nu/Nu* mice resulted in infiltrative carcinoma comprised of poorly defined and invasive glands, while a prominent desmoplastic reaction was observed in PanIN-like structures and PDAC. Boj et al. used these organoids to also identify novel genes that were upregulated in human PDAC, showing for the first time that they can serve as a tool for the discovery and dissection of pathways driving human pancreatic carcinogenesis. This could be rather helpful in the development of novel therapeutic and diagnostic approaches. A few months later, Muthuswamy’s lab reported the second successful attempt to develop pancreatic cancer tumoroids [45]. Fresh tissues of primary tumors were digested from patients and re-suspended initially in a special culture medium, developed by the authors, and called PTOM (progenitor and tumor organoid medium), and plated on a bed of Matrigel. As shown in Table 1, the group followed a rather complicated system to maintain, expand, and passage tumoroids, and their culture medium was mainly supplemented with B27, ascorbic acid, insulin, hydrocortisone, fibroblast growth factor 2, all-trans retinoic acid, and Y267632 [a cell-permeable, highly potent and selective inhibitor of Rho-associated, coiled-coil containing protein kinase (ROCK) inhibitor] without any stimulants of Wnt signaling (unlike in the Boj et al. method). The success rate was high (17 out of the 20 samples resulted in organoids, reaching 85% of success rate, as in the Boj et al. method), and tumor organoids showed similar morphological and cytological features, as well as patterns of differentiation markers, as the primary tumors they were derived from. All samples could be kept in long-term cultures, frozen, and thawed to re-establish cultures while maintaining phase and morphology across passages. Tumoroids could also generate tumors in xenograft models when injected at low inoculation numbers (50,000 cells) into NSG mice within a period of 4 to 7 weeks. Importantly, the developed tumors were reported to maintain the histoarchitectures present in the primary tumors from which the organoids originated and could also be used to re-establish organoid cultures. Following these remarkable studies, Walsh et al. reported in 2016 a simpler approach to develop human pancreatic cancer organoids (Table 1) [46]. They used a simple medium consisting of RPMI supplemented with 10% fetal bovine serum, 1% penicillin/streptomycin, and epidermal growth factor receptor at a concentration of 10 ng/mL. Authors reported the development of xenografts from just one patient with poorly differentiated pancreatic ductal adenocarcinoma. The sample was taken from a primary tumor biopsy during a Whipple procedure. Organoids were generated by mechanical dissociation with surgical scissors and a scalpel. The authors do not mention digestion with enzymes, passage of organoids, storage, subculture, or long-term self-renewal potential, nor animal testing to assess their ability to develop tumors. Although they do report that organoids were heterogeneous in their morphology, however; they did not compare them to the primary tumor to assess whether organoids recapitulate the morphological and cytological features of the tumor of origin. The authors cultured these organoids for three days and then used them for drug testing and optical imaging. Based on the information provided in this manuscript, it is difficult to understand whether these 3D cultures are real tumoroids or merely spheroids. In an extraordinary publication in 2018, the group of Kuo reported the development of organoids from 100 patients representing 19 distinct tissue sites and 28 unique disease subtypes using an improved approach of the ALI method of Ootani et al. [40,47]. In this work, authors reported that organoids could be formed even when using media supplemented with fetal calf serum alone; however, the best outcome was reached when the culture medium was supplemented with WNT3A, EGF, NOGGIN, and RSPO1 (WENR), which was used to expand and serially passage mechanically processed tumor fragments as ALI organoids (Table 1). These organoids could be further passaged with a high success rate (73%) and cryopreserved at a very high success rate of 80%. Notably, following this approach, authors were able to preserve both the non-immune and the immune stromal elements, which suggests that these organoids very closely mimic the real tumor. A drawback of this approach was that upon long term culture, i.e., over 4 passages of over 100 days in culture, some of the organoids did not retain the initial complex tissue architecture, which suggests that the PDOs developed using this method should carefully be monitored to ensure their reliability compared to the parental tumor.

A different approach for the development of organoids was reported by Romero-Calvo et al. [22]. In this study, researchers dissociated cells from biopsies and PDX in a two-step procedure. First, they used collagenase XI and dispase, and subsequently TrypLE^®^ Express (Thermo Fisher Scientific, Waltham, MA, USA, #12605-010) and DNAse I. To develop organoids, dissociated cells were embedded in growth factor–reduced (GFR) Matrigel and cultured in a basal medium called Intesticult (Stemcell Technologies, #6005), supplemented with: A83-01, fibroblast growth factor 10 (FGF10), gastrin I, N-acetyl-L-cysteine, Nicotinamide, B27 supplement, primocin and Y-27632 (Τable 1). These organoids were passaged weekly in a 1:2 subculturing ratio via mechanical dissociation with TrypLE^®^ Express. The success rate was 100%, as all 10 specimens (five directly from patients’ biopsies and another five from PDX) developed organoids. Subsequent H&E histological analysis showed that a high degree of correlation was maintained between the morphologic structure of the primary tumors and the corresponding organoids. For the PDX-derived organoids, although these organoids recapitulated the microscopic features of the primary tumors, their architecture was more complex and atypical compared with the organoid models, in part due to the presence of mouse stroma within the tumor, as authors reported. Analysis of the differential gene expression in organoids from one patient also revealed a high overlap, close to 90%, with the primary tumor. Thus, the overall results of this study indicated strong concordance at the morphological and molecular levels.

## 5. Tumor Organoids in Precision Medicine for PDAC

The first work of anticancer drug testing using tumoroids took place at Muthuswamy’s lab [45]. The first test they did was to analyze the response of tumoroids from five independent patient tumors to the standard drug for PDAC, gemcitabine. According to their report, all organoid tests showed a poor response to the drug, with an average growth inhibition of 30%. Unfortunately, they did not correlate the responses of these “personalized” organoids with that of the patients, as all patients had undergone surgery and were disease-free at the time of the study. The group proceeded to use the same five personalized organoids in a drug-screening approach to test five different epigenetic inhibitors. In a well-designed experiment, they first tested these inhibitors against non-tumor pancreatic organoids for toxicity, and the two least toxic were further tested against the tumoroids as a monotherapy or in combination with gemcitabine. Through this approach, they identified at least one inhibitor that resulted in a dose-dependent decrease in the proliferation rate of four out of the five tumoroids when combined with gemcitabine. The most active inhibitor was found to be the UNC1999, an inhibitor of EZH2 [a ‘writer’ of trimethylation of histone H3 at Lys27 (H3K27me3)]. Interestingly, all sensitive tumoroids, as well as the matching tumors and non-tumor organoids, were positive for the H3K27me3 mark, suggesting a strong correlation between the organoids and the primary tumor. It would have been of great interest to see how these studies would correlate with *in vivo* studies in NSG xenografts developed with the use of these organoids, but the group did not perform these experiments. Authors also report that tumor organoids retained patient-specific traits, such as repressive epigenetic marks, oxygen consumption, and EZH2 dependence, observations that underline the usefulness of their system for further drug screening. Walsh et al. [46] also used the organoids they developed to test their response in gemcitabine and AZD1480 (a novel ATP-competitive JAK2 inhibitor) alone or in combination. Organoids were treated for 24 h with gemcitabine, AZD1480, and their combination. As they report, while a significant reduction in the OMI index (optical metabolic imaging, used to evaluate the effect on organoids proliferation) was detected with gemcitabine and their combination, AZD1480 alone failed to induce a significant reduction. Additionally, the authors did not report any correlation between patients’ treatment and response. In this direction, Tiriac et al. attempted to correlate tumor organoid drug response to patients’ clinical outcomes. They developed a cohort of 66 PDAC patient-derived organoids and established a drug testing platform, suggesting that drug testing in PDO cultures could be used for treatment selection in patients within a clinically relevant timeframe. The organoid response to drug testing, termed pharmacotyping, was performed on PDOs treated with 5 common clinically used chemotherapeutic agents: gemcitabine, nab-paclitaxel, irinotecan (SN-38), 5-fluorouracil (5-FU) and oxaliplatin. The authors collected retrospective clinical follow-up from 9 patients and reported that eight out of nine (89%) organoids responded similarly to the corresponding patients. More specifically, 6 patients treated with at least one drug to which PDOs were found to be sensitive showed an improved progression-free survival (PFS), whereas two out of three patients treated with a drug to which the corresponding PDOs were resistant rapidly progressed. Authors suggested that the PDOs’ response to drug screening parallels patients’ sensitivity profiles to chemotherapy [51]. The authors further evaluated the efficacy of other targeted agents as potential anticancer drugs for PDAC using these PDOs. Targeted agents like selumetinib (MEK inhibitor), afatinib, everolimus, and LY2874455 (fibroblast growth factor receptor inhibitor) were tested. Authors found that PDOs harboring ERBB2 amplifications and EGFR mutations were sensitive to the tyrosine kinase inhibitor afatinib, whereas a PDO carrying an oncogenic PIK3CA allele was sensitive to the mTOR agent everolimus. A similar correlation between clinical response and gemcitabine sensitivity was reported by Driehuis et al. [52]. The authors exposed 24 established pancreatic PDOs to an extensive drug screening with 76 chemotherapeutic agents aiming to assess PDOs’ drug sensitivity and correlate it to the patients’ clinical outcomes. However, clinical data were available only for 4 patients and all of them were treated with gemcitabine. Authors reported that one patient, with the corresponding PDO being resistant to gemcitabine, developed distant metastasis during treatment, while another patient with a PDO sensitive to gemcitabine had stable disease and a decrease in liver metastasis after gemcitabine treatment. The corresponding PDOs from the two other patients showed an intermediate response in the *in vitro* assays, and in accordance with this finding, the two patients showed stable disease during gemcitabine treatment. Based on these findings, authors suggest that the *in vitro* sensitivity of the PDOs to gemcitabine correlates to patients’ clinical response. However, it should be noted that the patients’ cohort and the number of corresponding PDOs in this study were relatively small [52]. In the same study, authors used PDOs to evaluate the potential of a new targeted agent, EZP01556, as an anticancer agent. This inhibitor specifically targets the arginine methyltransferase 5 (PRMT5) protein. PRMT5 is reported as a synthetic lethal gene in methylthioadenosine phosphorylase deficient (MTAP^−^) cells [53].

In their study, Romero-Calvo et al. [22] reported the testing of organoids for dosage-dependent and drug-specific responses. They studied the response of organoids from two patients to gemcitabine and a combination of gemcitabine and abraxane. The authors reported a dose-response relationship to gemcitabine in both cases; however, this did not reflect the patients’ clinical outcome, as both patients were unresponsive to gemcitabine. Organoids from one patient were also tested with a combination of gemcitabine plus abraxane and showed a very good response consistent with the corresponding PDX. Nevertheless, this could not be correlated with clinical data since this patient did not receive this regimen. However, the key finding of the study is the isolation of two clones from the organoids, which underline the potential of these models to provide valuable information about clonal populations. Another study from Seppälä et al. [54] aimed to determine whether the PDOs’ pharmacotyping could guide the patients’ postoperative chemotherapeutic selection within a clinically relevant time frame. Authors first studied PDOs’ drug sensitivity in early-, as compared to late-passage organoids derived from the same primary tumor. Interestingly, they found that the established clones showed a similar pharmacotyping profile, which was consistent across over a dozen of samples (7–89 days between passages). Since they confirmed the PDOs’ pharmacotyping stability, authors reported the days in culture before pharmacotyping for 13 PDOs and found that they managed to complete the drug screening within a range of 18–102 days (median 48 days). Considering the median time between surgery and the initiation of chemotherapy, which was 62 days, authors suggest that rapid PDOs’ pharmacotyping could be a feasible strategy with clinical relevance in PDAC treatment. However, once again in this study, authors did not report any correlation between patients’ response and outcome [54]. In another study, Farshadi et al. [55] developed 5 PDOs from pancreatic cancer patients who has undergone eight cycles of the FOLFIRINOX regimen and 5 PDOs from treatment-naive patients. The authors studied the PDOs’ drug response to oxaliplatin, 5-FU, and SN38 as monotherapies. The results suggest that the clinical drug response for three out of five treatment-naive patients, who received FOLFIRINOX treatment, mirrored the *in vitro* drug sensitivity of their corresponding PDOs. Interestingly, one out of five FOLFIRINOX-treated PDOs showed higher sensitivity to oxaliplatin *in vitro* compared to the others, and computerized tomography of this patient’s tumor showed that this was the only patient with a partial response (PR) to FOLFIRINOX. Congruently, one out of five treatment-naive PDOs showed less sensitivity to different concentrations of FOLFIRINOX *in vitro*, mirroring the observed clinical resistance of this patient to adjuvant FOLFIRINOX. However, these results have been observed in a very small number of patients to draw definite conclusions and must be validated in a larger cohort. In a more recent study, Grossman et al. [56] established 12 PDOs suitable for drug testing from 12 patients with pancreatic cancer enrolled in the study. PDOs sensitivities to various drugs such as gemcitabine, 5-FU, oxaliplatin, SN-38, and paclitaxel were tested. The drugs tested in each PDO were selected according to the treatments each patient received or was likely to receive. Researchers calculated the area under the curve (AUC) for each drug on each PDO as an index for the clinical response prediction, and they observed a high degree of correlation between clinical and PDO responses. For example, in one patient, the developed PDO showed the highest degree of sensitivity to irinotecan, followed by trametinib, and in accordance with this finding, the patient from whom this PDO was derived showed stable disease (SD) on the agent FOLFIRINOX (contains irinotecan) and partial response (PR) on trametinib/lapatinib, suggesting that the AUC values calculated from the PDOs could provide useful insights into the clinical responses to treatment [56].

## 6. Discussion

In cancer research, human cell lines, mouse models, and patient-derived xenografts (PDXs) have been traditionally used as the main experimental research platforms. However, extrapolating results derived from these cancer models to patients has become a major bottleneck, especially in the drug discovery process. Patient-derived organoids (PDOs) have recently emerged as a powerful model for precision medicine applications. Currently, there is an increasing number of organoids used in xenograft formation and molecular profiling, and patient-derived organoid models are extensively used as *in vitro* screening platforms with the potential to predict the best therapeutic options for individual patients. Numerous studies have been performed in the field of PDO drug screening; however, it remains unclear whether and how this approach could match the patients’ clinical responses and, more importantly, whether these results can be translated into a clinical decision strategy. 

PDOs are establishing their place in the effort to bridge the gap between animal models and human patients, as multiple studies are currently aiming to optimize their use, focusing on their benefits and their potential applications. Several benefits of the PDOs over previous models lie in their relative rapidness, robustness, high success rate, reproducibility, and affordability. Animal models, which are still the most used preclinical models, require much longer times to be developed and established compared to organoids, which usually can be established within a few weeks. However, the time required for the organoids’ establishment might vary greatly. There are studies where PDOs were established from 4 up to 33 days [57] or within a range of 18–102 days [54] showing that the time-consuming aspect of organoids’ development is a parameter that cannot be overlooked.

The success rate of organoids is an issue that needs to be further addressed. Although the reported rate of organoid development is high (70–73%) [45], it might vary depending on the tumor type or the amount of available material, which is usually limited. Boj et al., [44] report that they failed to establish organoids from a patient who underwent neo-adjuvant chemotherapy due to extensive necrosis in the sample, as revealed by histological examination. They also failed to establish organoids from a patient sample highly enriched in stromal cells, and thus tumor cells were not enough for the formation of the tumor organoid. Similarly, Huang et al. managed to isolate 17 organoids from 20 PDAC patient samples. The three samples that didn’t lead to organoid formation were poorly to moderately differentiated PDAC and intraductal papillary mucinous cystic neoplasms [45]. The methods used for tumor sample acquisition (i.e., biopsies vs surgical resections) can also impact the efficacy of organoid generation. In the work of Driehuis et al., it is reported that the generation of organoids from biopsies was limited to 31%. The authors also concluded, through whole-genome sequencing (WGS) of the generated organoids, that specific genes, such as SMAD4, and their mutation status can influence organoids’ growth in specific media. Therefore, they suggest testing samples to identify appropriate organoid culture media, as specific growth factors may be required for the establishment of patient derived organoids [52]. Indeed, although the culture of organoids is generally straightforward, the issue of culture conditions is critical and requires proper consideration. Various protocols from different groups exist, resulting in a variety of outcomes due to the lack of a widely accepted standardized methodology (see Table 1 and Table 2). This raises concerns regarding the validity of the models and increases the difficulty of comparing the results across different studies and research groups.

The maintenance of organoids remains an unresolved challenge within the scientific community. In a recently published study by Mäkinen et al., authors report a 100% success rate in organoid formation, yet only approximately 50% of the organoids could be cultured beyond passage three. Storage of organoids towards developing a biobank for future use doesn’t seem to be quite successful, as the success rate for re-establishing stored organoids is relatively poor, not exceeding 50% [58]. These limitations pose significant obstacles to the widespread use of organoids in precision medicine.

Another consideration regarding organoid models is the lack of inter-organ communication. Organoids are limited to replicating tissue-specific physiology and to overcome this limitation researchers started to focus on the development of models able to connect multiple organoids and study the communication between them. For example, in a recently published study, researchers studied the communication between pancreas, liver, and gastrointestinal organoid systems derived from human pluripotent stem cells (PSCs), and they managed to develop a functionally and structurally integrated organoid system [59]. In that direction, it would be of great interest and usefulness the development of models that will offer the ability to study multi-organ communication *in vitro*, aiming ultimately towards personalized therapeutic approaches.

In pancreatic cancer research, organoids are currently studied as model systems for the development of novel precision medicine strategies. Several studies aim to evaluate patient-derived organoids as predictive platforms for the choice of chemotherapy regimens and entirely novel drugs and combinations for the treatment of PDAC. Researchers aim to develop pancreatic human organoids that could become a tool for rapid drug assessment of the chemosensitivity of individual tumors (i.e., patients) before or in parallel with implementing treatment in patients with PDAC. Currently, there are very few studies that correlate pancreatic PDOs’ drug screening results to patients’ clinical responses, as we summarized above. In these studies, researchers have shown that PDOs’ response to drug testing was able to parallel patients’ clinical response [52,55,56,60]. However, these studies have been performed on relatively small cohorts, highlighting the importance of well-planned, more extensive co-clinical trials in larger patient cohorts. 

Another important aspect to consider is that organoids are mainly constituted of epithelial cells and lack numerous tumor microenvironment (TME) elements, including different types of cells such as fibroblasts, endothelial, immune, and other stromal cells. This is an important limitation of the organoid model system, and efforts have been made to establish co-cultures of organoids with other cell types in order to generate an advanced model able to recapitulate tumor heterogeneity, structure, and putative cell-to-cell interactions. In cancer, it is well known that the tumor microenvironment including extracellular matrix (ECM), stromal elements, vascular structures as well as infiltrating immune cells is of critical importance for tumor progression and metastasis. Also, many drug responses on human tumors are related and eventually strongly affected by the tumor microenvironment elements. In that context, several variations to PDO cultures are under development, and the tumor stroma and immune system are being incorporated in an increasing number of studies. For example, in a recent study, Tsai et al. reported the co-culture of pancreatic cancer organoids with patient derived CAFs and peripheral blood lymphocytes to assess lymphocyte migration and the activation status of CAFs [61]. Organoid vascularization is another aspect that should be considered. Organoids lack vascular circulation, and to overcome this limitation, researchers have been co-culturing organoids with endothelial cells in microfluidic devices [62]. Nevertheless, the maintenance of different cell populations over time and the long-term preservation of stromal components and immune cells in the organoid culture are still under investigation. The culture medium composition needs to be optimized to support the diverse cell types while avoiding clone selection. The significance of these specific issues is underscored by the work of Raghavan et al. in which authors clearly show that the *ex vivo* TME and the media formulation may not only drive the state of the pancreatic cancer cells grown as PDOs in a non-genetic manner, but also distinctly affect the drug responses. In this work, researchers, by simply omitting or including TGF-β in the culture medium, managed to shift the sensitivity of the tested organoids from agents targeting DNA-damage repair pathways to inhibitors of the mitogen-activated protein kinase (MAPK) pathway [63].

Given the detailed discussion above, it is evident that organoids are currently not suitable to support clinical decision-making. The extended time required to establish organoids, the moderate success rate of their development, the dependence on the methodology used to obtain the material, the challenges in culturing, maintaining, and storing them for future use, the lack of inter-organ communication, and the lack of numerous tumor microenvironment elements remain significant barriers that the scientific community must overcome for organoids to find their place in the clinical setting.

Efforts are currently underway to overcome these limitations. In 2021, Osuna de la Pena et al. reported the use of a non-Matrigel based ECM, much easier to be standardized. This ECM was based on hydrogels developed by amphiphilic polypeptides that could be self-assembled and supplemented with multiple PDAC ECM components like collagen type I, fibronectin, laminin, hyaluronan, and components of the TME like patient-derived pancreatic stellate cells (PSCs) and primary macrophages. The subsequent testing of PDOs developed in this matrix revealed that the responses of the organoids against nab-paclitaxel and gemcitabine were correlated with these of the corresponding patients. Although this testing was limited to only three organoids, it is an essential step forward in addressing issues related to organoid culture [64]. Recent advances in technology aiming to merge the fields of organoid research and organ-on-a-chip research are also particularly exciting. The aim is the development of an ‘organoid-on-a-chip’ technology by integrating human organoids with organ-on-a-chip engineering. This could provide a superior *in vitro* microfluidic platform, that will enable the separate culture of distinct organoid types, permitting their communication, for preclinical drug screening [65]. However, these models are expensive, still largely under development, constantly being optimized and improved, and difficult to use for high-throughput drug screening experiments and obviously for the purposes of precision medicine. 

Despite the remaining challenges, human organoids represent a dynamically developing technology with enormous potential in the field of basic and clinical translational research. Numerous studies underscore this potential. The investigation of mechanisms that result in drug resistance and strategies for overcoming them is a field of immense use of PDOs. Ponz-Sarvise et al., in an effort to study and overcome the resistance observed in PDAC upon combined MAPK and PI3K inhibition, identified the overactivation of ERBB as a critical parameter in this pathway. They further demonstrated that the combined MEK and ERBB inhibition using the MEK inhibitor selumetinib and the pan-ERBB inhibitor neratinib of human organoid orthotopic xenografts resulted in tumor regression in short-term intervention studies [66]. Another example is the investigation and comprehension of the underlying mechanism of resistance development of PDAC against FOLFIRINOX regimen, as demonstrated in a recent study by Bachir et al., with the aid of PDOs [67]. Besides the use of PDOs for drug studies, these models are already in use to uncover and validate novel targets and signaling pathways involved in pancreatic cancer progression [51,52,56,68]. Organoids are extensively used to shed light on the mechanisms driving the metastatic potential of PDAC. With the aid of PDOs, Huang et al., identified SMAD4 as the transcription factor responsible for promoting EMT and orchestrating the invasion program in PDAC [69].

Their use in the development of more clinically relevant animal models of cancer is another field where organoids are gaining ground. PDO xenografts are reported to better represent the cancer cell state of the parental tumor, and thus serve as more effective models for drug testing. They can be easily genetically modified, allowing for broad applications in basic cancer research and show a higher success rate as compared to PDX models at a more reasonable time and cost (for a more comprehensive review on this see [70]).

Precision medicine is an approach to medical treatment that considers individual variability in genes, environment, and lifestyle for each person to tailor the most effective treatment and prevention strategies for that individual. PDOs are 3D cell culture systems that can replicate the structure and function of specific organs and tumors to a significant extent while retaining individual gene signatures, allowing researchers to study disease development and drug responses in a more physiologically relevant environment.

Considering the rapid technological and methodological advances in the field, we believe that human organoid systems have the potential to guide personalized therapeutic approaches. Future advancements are anticipated to achieve a robust cancer model system and a powerful tool not only for understanding cancer biology, biomarker research and drug screening but also able to guide clinical decisions, improve cancer chemotherapies, and become a cornerstone for a more precise oncology approach. 

## Figures and Tables

**Figure 1 biomedicines-11-00890-f001:**
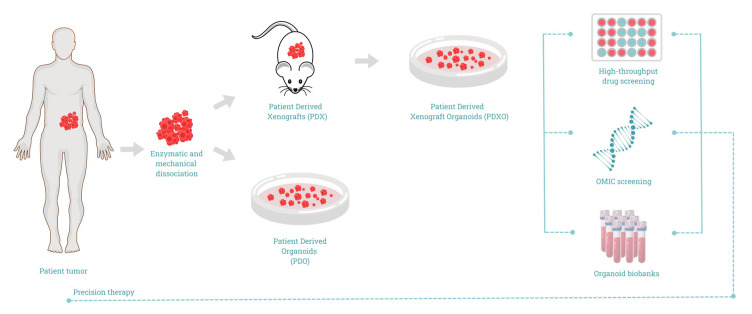
Schematic representation of patient-derived organoids (PDO) development for precision cancer medicine.

**Table 1 biomedicines-11-00890-t001:** Human PDAC organoid culture methodology.

Reference [No]	Sample	Matrix	Culture Medium	Transplantation of Organoids	Success of Organoid Culture	Passage	Targeted Treatment on Organoids	Storage
Boj et al. 2015 [44]	Fresh tumors from surgical resection and FNA biopsies	GFR Matrigel (embedded)	AdDMEM/F12 supplemented with: (1) Glutamax 1X, (2) Hepes 1X, (3) Noggin 10% *v*/*v* or recombinant protein 0.1 μg/mL, (4) Gastrin 10 nM, (5) Nicotinamide 10 mM, (6) R-spondin-1 10% *v*/*v*, (7) EGF 50 ng/mL, (8) FGF-10 100 ng/mL, (9) N-acetyl-L-cysteine 1 mM, (10) B27 supplement, (11) Wnt3a 50% *v*/*v*, (12) primocin 1 mg/mL, (13) Penicillin-streptomycin 1X, (14) A83-01 0.5 μΜ	PDOX creation by orthotopical injection of human PDAC organoids to *Nu/Nu* mice, PanIN like lesions	75%	Indefinite passaging capability	Not mentioned	Organoids can be cryo-preserved
Huang et al. 2015 [45]	Fresh tumors from surgical resection	Bed of Matrigel	Pancreatic Progenitor and Tumor Organoid Media (PTOM) consisted of DMEM with (1) 1% B27, (2) 50 ug/mL ascorbic acid, (3) 20 μg/mL insulin, (4) 0.25 μg/mL hydrocortisone, (5) 100 ng/mL FGF2, (6) 100 nM all-trans retinoic acid, (7) 10 μΜ Υ267632. At day 8 in 3D culture, culture medium was replaced with fresh pancreatic organoid maintenance medium (POMM, which contains 1% B27 and 50 μg/mL ascorbic acid) with 5% Matrigel. Culture media was replaced every 4 days. Pancreatic Organoid Differentation Media 1 (PODM 1) consisted of DMEM supplemented with (1) 1% B27, (2) 300 μΜ 2-phospho ascorbic acid, (3) 100 ng/ml FGF7, (4) 10 ng/mL EGF, (5) 1 μΜ A8301 and (6) 1 μΜ DBZ. Pancreatic Organoid Differentation Medium 2 (PODM 2) consisted of DMEM supplemented with (1) 1% B27, (2) 300 μΜ 2-phospho ascorbic acid, (3) 100 ng/mL FGF7 and (4) 10 ng/mL EGF.	0.3–0.5 million cells were injected into the mammary glands of female NOD/SCID mice	85% (17/20 samples)	At day 16, organoids are treated with collagenase for 2 hours and then additionaly with trypsin for another 10–30 min. The cells are collected and re-seeded in 3D culture.	Drugs targeting epigenetic regulators were tested, i.e A366 and UNC1999, alone or in combination with Gemcitabine	Tumor organoids could be freezed-thawed and still re-establish cultures that preserve phase and hematoxylin and eosin stain (H&E) morphology across passages
Walsh et al. 2016 [46]	Biopsy from a patient with poorly differentiated pancreatic ductal adenocarcinoma	Matrigel (embedded at 1:2 ratio)	RPMI supllemented with (1) 10% FBS, (2) 1% Penicillin/Streptomycin, (3) 10 ng EGFR. Culture medium was replaced every 3 days.	Not reported	Not reported	Not mentioned	Organoids were cultured for 3 days and treated with (1) DMSO (control), (2) Gemcitabine (25 μg/mL), (3) AZD1480 (100 nM), (4) AZD6244 (4 μM), (5) XL147 (25 nM) and (6) combination of gemcitabine plus (a) AZD1480, (b) AZD1480 + AZD6244, (c) AZD1480 + AZD6244 + XL147	Not reported
Romero-Calvo et al. 2019 [22]	Biopsies (n = 10) from PDAC patients and PDX derived organoid cultures	GFR Matrigel (embedded)	Complete media [Intesticult (Stemcell Technologies, 6005), (1) A83-01 (0.5 mmol/L), (2) fibroblast growth factor 10 (FGF10, 100 ng/mL), (3) gastrin I (10 nmol/L), (3) N-acetyl-L-cysteine (10 mmol/L), (4) nicotinamide (10 mmol/L), (5) B27 1X, (6) primocin (1 mg/mL), and (7) Y-27632 (10.5 mmol/L)	Not reported	100%	Organoids were passaged via mechanical dissociation with TrypLE Express and passage was performed weekly with a 1:2 ratio.	Chemotherapeutic drugs tested: (1) gemcitabine (3 nmol/L, 10 nmol/L, 30 nmol/L) (2) FOLFIRINOX (10 nmol/L) (3) Abraxane (10 nm/L)	Not reported
Neal et al. 2018 [47]	100 patients representing 19 distinct tissue sites and 28 unique disease subtypes	ALI method	ADMEM/F12 supplemented with 50% Wnt3a, RSPO1, Noggin-conditioned media (L-WRN) with HEPES (1 mM), Glutamax (1X), Nicotinamide (10 mM), N-Acetylcysteine (1 mM), B-27 without vitamin A (1X), A83-01 (0.5 mM), Pen-Strep Glutamine (1X), Gastrin (10 nM), SB-202190 (10 mM), and EGF (50 ng/mL). The transwell containing tumor tissue and collagen was placed into an outer 60 mm cell culture dish containing 1.0 mL of medium.	Not reported	73% (at 1-month culture across tumor histologies)	After continued growth (>4 passages, >100 days) some PDOs did not maintain the complex tissue architecture exhibiting a simple, cystic morphology. PDOs could be xenografted into immunocompromised mice and re-derived as organoids.	Not reported	80% (cryo-recovered and serially re-propagated every few weeks)
Choi et al. 2019 [48]	Biopsies of liver metastasis from patients with PDAC were used to develop PDOX in female *Hsd:athymic nude-Foxn1* mice aged 5–8 weeks. Then, first passage cells isolated from F1 PDX tissues were used to develop organoids.	GFR Matrigel	Advanced DMEM/F12 with (1) B27, (2) N-acetylcysteine, (3) EGF, (4) FGF-10, (5) R-spondin 1 and (6) Noggin. Medium was replaced every 3 days.	Not reported	Not mentioned	The researchers used first passage cells isolated from F1 PDOX tissues	Organoids were treated with (1) DMSO (control), (2) Gemcitabine-HCL and (3) Albumin-bound paclitaxel for 7 days	Not mentioned
Hennig et al. 2019 [11]	Human primary tumor samples from 31 treatment-naïve PDAC patients: 25 specimens from surgical tumor resections and 6 from endoscopic ultrasound- (EUS-) guided fine needle aspiration (FNA)	GFR Matrigel	Advanced DMEM/F12 suplemented with (1) WNT3a 50% *v*/*v*, (2) noggin 10% *v*/*v*, (3) Rspondin1 10% *v*/*v*, (4) B27 1X, (5) nicotinamide 10 mM, (6) Gastrin 1 nM, (7) n-acetyl-L-cysteine 1 mM, (8) primocin 1 mg/mL, (9) mEGF 50 ng/mL, (10) hFGF10 100 ng/mL, (11) A83-01 0.5 μΜ, (12) Ν2 1Χ	Not reported	71% (68% from surgical resections and 83% from fine needle aspirations)	Not mentioned	Chemotherapeutic drugs tested: (1) Gemcitabine at 1 μΜ, 200 nM, 100 nM, 50 nM, 25 nM, 10 nM and 1 nM, (2) 5-Fuorouracil at 50 μΜ, 25 μΜ, 10 μΜ, 5 μΜ, 1 μΜ, 100 nM and 10 nM, (3) Oxaliplatin at 250 μΜ, 25 μΜ, 10 μΜ, 1 μΜ, 100 nM, 10 nM and 1 nM and (4) Irinotecan at 250 μΜ, 25 μΜ, 10 μΜ, 1 μΜ, 100 nM, 10 nM and 1 nM	Repeated freeze-thaw cycles

**Table 2 biomedicines-11-00890-t002:** Supplements used for human pancreatic organoid culture and their function.

Supplements	Function
2-phospho ascorbic acid	Maintains cellular differentiation
A 83-01	TGF-b inhibitor
B27 supplement	Vitamins and growth factors mix
EGF	Mitogen.
EGFR	Mitogen
FGF-10	Stimulates cell proliferation
FGF-2	Stimulates cell proliferation
FGF-7	Stimulates cell proliferation
Gastrin	TGF-b inhibitor
Glutamax	Substitute for L-glutamine
HEPES	Provides a buffered pH environment
IGF-1	Mitogen
N-acetyl-L-cysteine	Precursor in glutathione synthesis
N2 supplement	Vitamins and growth factors mix
Nicotinamide	Maintain cell stemness and cystic phenotype
Noggin	Inhibits BMP-4
Penicillin/streptomycin	Avoid bacterial contamination
PGE2	Maintain cell stemness and cystic phenotype
Primocin	Antibiotic
RSPO-1	Wnt signaling activator
SB202190	p38 MAPK inhibitor
Wnt3A conditioned Medium	Wnt signaling activator
Y-27632	ROCK inhibitor to prevent anoikis

## Data Availability

Not applicable.

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
