# Peer review of "Pancreatic Cancer Organoids: An Emerging Platform for Precision Medicine?"

_biomedicines, 2023, doi:10.3390/biomedicines11030890_

Round 1
Reviewer 1 Report
This review describes the pancreatic cancer organoids for precision medicine. The manuscript presents deficiencies that should be revised to reach the possibility of publication.
1. Describe the disadvantages or failure cases of pancreatic cancer organoids for precision medicine.
2. Explain the difference with the paper Pancreatic Cancer Organoids in the Field of Precision Medicine: A Review of Literature and Experience on Drug Sensitivity Testing with Multiple Readouts and Synergy Scoring (Cancers (Basel). 2022 Feb; 14(3): 525.)
Other comments:
Human PDAC organoid culture methodology were described in detail.
Yes, conclusions are consistent with evidence and references are appropriate.
Author Response
We thank the reviewer for the effort she/he put into reviewing our manuscript and the constructive comments that helped us to improve and strengthen our manuscript. Please find below our responses to the reviewer’s specific comments:
This review describes the pancreatic cancer organoids for precision medicine. The manuscript presents deficiencies that should be revised to reach the possibility of publication.
- Describe the disadvantages or failure cases of pancreatic cancer organoids for precision medicine
We now have included the following section in the discussion part (lines 487-526):
"The success rate of organoids is a further issue that needs to be addressed. Although the reported rate of organoid development is high (70-73%) [44], it might vary depending on the tumor type or the amount of available material, which is usually limited. Boj et al., [41] report that they failed to establish organoids from a patient who underwent neo-adjuvant chemotherapy due to the occurrence of extensive necrosis in the sample, as revealed by histological examination. They also failed to establish organoids from a patient sample highly enriched in stromal cells and thus tumor cells were not enough for the formation of the tumor organoid. Similarly, Huang et al., managed to isolate 17 organoids from 20 PDAC patient samples. The three samples that didn’t lead to organoid formation were poorly- to moderately differentiated PDAC and intraductal papillary mucinous cystic neoplasms [44]. The methods used for tumor sample acquisition (i.e., biopsies vs surgical resections) can also impact the efficacy of organoid generation. In the work of Driehuis et al., it is reported that the generation of organoids from biopsies was limited to 31%. The authors also concluded, through whole-genome sequencing (WGS) of the generated organoids, that specific genes, such as SMAD4, and their mutation status can influence organoids’ growth in specific media. Therefore, they suggest testing samples to identify appropriate organoid culture media, as specific growth factors may be required for the establishment of patient derived organoids [48]. Indeed, although the culture of organoids is generally straightforward, the issue of culture conditions is critical and requires proper consideration. Various protocols from different groups exist, resulting in a variety of outcomes due to the lack of a widely accepted standardized methodology (see tables 1 and 2). This raises concerns regarding the validity of the models and increases the difficulty of comparing the results across different studies and research groups.
The maintenance of organoids remains an unresolved challenge within the scientific community. In a recently published study by Mäkinen et al., authors report a 100% success rate in organoid formation, yet only approximately 50% of the organoids could be cultured beyond passage three. Storage of organoids towards developing a biobank for future use doesn’t seem to be quite successful as the success rate for re-establishing stored organoids is relatively poor being not higher than 50% [56]. These limitations pose significant obstacles to the widespread use of organoids in precision medicine."
- Explain the difference with the paper Pancreatic Cancer Organoids in the Field of Precision Medicine: A Review of Literature and Experience on Drug Sensitivity Testing with Multiple Readouts and Synergy Scoring (Cancers (Basel). 2022 Feb; 14(3): 525.)
This is a very nice and interesting paper indeed and we thank the reviewer for bringing this work to our attention. This is a mixed manuscript combining both a literature review and the original research work of the authors. Some of the information in the research part of this work is, indeed, essential and is now included in our manuscript.
Regarding the review part of this manuscript, we believe that our manuscript provides more in-depth information regarding the potential use of these platforms as tools for clinical decision-making. We also provide a detailed methodology for pancreatic organoids development of the supplements (part 3.1 and tables 1 and 2) which could be used as an experimental guide as well. We further provide a detailed description of the works regarding the use of these platforms for drug testing and specifically of works related to precision medicine with new information to be included as well (see ref 50). Finally, we extensively discuss the pitfalls of this technology that hinder their use in the clinics till now but the progress as well towards addressing some of these concerns. We believe that our manuscript focusing solely on the potential use of organoids in precision medicine and as generic tools for drug discovery provides the reader a comprehensive image of the field and mainly of the needs to be addressed for these platforms to find their place in the clinics and ultimately, to improve clinical decision-making and cancer treatments.
We hope we have addressed all concerns properly. Thank you again!
Reviewer 2 Report
Query#1
In the paragraph “Introduction”: the authors properly reported pancreatic cancer incidence and mortality. However, in my opinion is important also to summarize the different hallmarks involved in pancreatic cancer progression and resistance. For instance, updated scientific reviews reported that several protein kinases are involved in pancreatic cancer resistance. Remarkably. the inhibition of kinases involved in the regulation of cell cycle such as cyclin dependent kinase 1 (CDK1), demonstrated to reduce pancreatic cancer growth in primary and immortalized pancreatic cancer cell lines.
At this purpose I suggest to the authors to cite the following paper:
Wijnen, R., Pecoraro, C., Carbone, D., Fiuji, H., Avan, A., Peters, G. J., Giovannetti, E., & Diana, P. (2021). Cyclin Dependent Kinase-1 (CDK-1) Inhibition as a Novel Therapeutic Strategy against Pancreatic Ductal Adenocarcinoma (PDAC). Cancers, 13(17), 4389. https://doi.org/10.3390/cancers13174389
Ding, L., & Billadeau, D. D. (2020). Glycogen synthase kinase-3β: a novel therapeutic target for pancreatic cancer. Expert opinion on therapeutic targets, 24(5), 417–426. https://doi.org/10.1080/14728222.2020.1743681
Pecoraro, C., Parrino, B., Cascioferro, S., Puerta, A., Avan, A., Peters, G. J., Diana, P., Giovannetti, E., & Carbone, D. (2021). A New Oxadiazole-Based Topsentin Derivative Modulates Cyclin-Dependent Kinase 1 Expression and Exerts Cytotoxic Effects on Pancreatic Cancer Cells. Molecules (Basel, Switzerland), 27(1), 19. https://doi.org/10.3390/molecules27010019
Edderkaoui, M., Chheda, C., Soufi, B., Zayou, F., Hu, R. W., Ramanujan, V. K., Pan, X., Boros, L. G., Tajbakhsh, J., Madhav, A., Bhowmick, N. A., Wang, Q., Lewis, M., Tuli, R., Habtezion, A., Murali, R., & Pandol, S. J. (2018). An Inhibitor of GSK3B and HDACs Kills Pancreatic Cancer Cells and Slows Pancreatic Tumor Growth and Metastasis in Mice. Gastroenterology, 155(6), 1985–1998.e5. https://doi.org/10.1053/j.gastro.2018.08.028
Query#2
In the paragraph 2 titled current status, line 88 page 3, please specify to which cell lines are referring the authors.
Query#3
The section entitled Discussion is well-formulated and full of information with a specific deepening on the potential of organoids as culture model to evaluate the specific chemosensitivity of PDAC patients. However, I suggest to the authors to divide the paragraph “Discussion” in two paragraphs, adding another section which summarize the several drawbacks of organoid cell cultures.
Author Response
We thank the reviewer for the effort she/he put into reviewing our manuscript, the encouraging words, and the constructive comments that helped us to improve and strengthen our manuscript. Please find below our responses to the reviewer’s specific comments:
1. In the paragraph “Introduction”: the authors properly reported pancreatic cancer incidence and mortality. However, in my opinion is important also to summarize the different hallmarks involved in pancreatic cancer progression and resistance. For instance, updated scientific reviews reported that several protein kinases are involved in pancreatic cancer resistance. Remarkably. the inhibition of kinases involved in the regulation of cell cycle such as cyclin dependent kinase 1 (CDK1), demonstrated to reduce pancreatic cancer growth in primary and immortalized pancreatic cancer cell lines.
At this purpose I suggest to the authors to cite the following paper:
Wijnen, R., Pecoraro, C., Carbone, D., Fiuji, H., Avan, A., Peters, G. J., Giovannetti, E., & Diana, P. (2021). Cyclin Dependent Kinase-1 (CDK-1) Inhibition as a Novel Therapeutic Strategy against Pancreatic Ductal Adenocarcinoma (PDAC). Cancers, 13(17), 4389. https://doi.org/10.3390/cancers13174389
Ding, L., & Billadeau, D. D. (2020). Glycogen synthase kinase-3β: a novel therapeutic target for pancreatic cancer. Expert opinion on therapeutic targets, 24(5), 417–426. https://doi.org/10.1080/14728222.2020.1743681
Pecoraro, C., Parrino, B., Cascioferro, S., Puerta, A., Avan, A., Peters, G. J., Diana, P., Giovannetti, E., & Carbone, D. (2021). A New Oxadiazole-Based Topsentin Derivative Modulates Cyclin-Dependent Kinase 1 Expression and Exerts Cytotoxic Effects on Pancreatic Cancer Cells. Molecules (Basel, Switzerland), 27(1), 19. https://doi.org/10.3390/molecules27010019
Edderkaoui, M., Chheda, C., Soufi, B., Zayou, F., Hu, R. W., Ramanujan, V. K., Pan, X., Boros, L. G., Tajbakhsh, J., Madhav, A., Bhowmick, N. A., Wang, Q., Lewis, M., Tuli, R., Habtezion, A., Murali, R., & Pandol, S. J. (2018). An Inhibitor of GSK3B and HDACs Kills Pancreatic Cancer Cells and Slows Pancreatic Tumor Growth and Metastasis in Mice. Gastroenterology, 155(6), 1985–1998.e5. https://doi.org/10.1053/j.gastro.2018.08.028
We thank the reviewer for bringing these works to our attention. We have now included the following section in the introduction part (lines 47-54)
"Pancreatic cancer progression and resistance are also complex processes involving multiple mechanisms, including genetic and epigenetic alterations, tumor microenvironment, metabolic reprogramming, immune evasion, and tumor heterogeneity. Recent studies showed that several protein kinases, like cyclin-dependent kinase 1 (CDK1) [17–20] play a significant role in pancreatic cancer resistance by modifying proteins involved in key signaling pathways. These modifications can lead to the activation of cellular processes that promote cancer cell growth, survival, and resistance to chemotherapy."
2. In the paragraph 2 titled current status, line 88 page 3, please specify to which cell lines are referring the authors.
Following the reviewer's query, we have added the following information in section "2. Current status" (lines 97-101)
"Such experiments are described in the paper by Keepers et.al., [18] where the authors compare two assays commonly used for drug screening, the tetrazolium (MTT) and the sulforhodamine B (SRB) assays, in two established cancer cell lines, the HT29, a colorectal cancer cell line and 11B, a squamous cell carcinoma cell line of the head and neck."
3. The section entitled Discussion is well-formulated and full of information with a specific deepening on the potential of organoids as culture model to evaluate the specific chemosensitivity of PDAC patients. However, I suggest to the authors to divide the paragraph “Discussion” in two paragraphs, adding another section which summarize the several drawbacks of organoid cell cultures.
Following reviewer's suggestion we now have added a new paragraph in the discussion section that reads as follows (lines 568-574)
"Given what has been discussed in detail above, it is obvious that organoids are not yet suitable to support clinical decision-making. The time required to establish organoids, the success rate of organoid development, the dependence on the methodology used to obtain the material, the challenges in culturing, maintaining, and storing of organoids for future use, the lack of inter-organ communication, and the lack of numerous tumor microenvironment elements remain significant barriers that the scientific community must overcome for organoids to find their place in the clinical setting."
We have as well extensively revised the manuscript to address linguistic errors
We hope that we have addressed properly all the concerns that the reviewer raised. Thank you again!
Reviewer 3 Report
Comments to authors:
This review demonstrates an overview of pancreatic cancer organoids for precious medicine. It systematically and explicitly illuminated the necessity, feasibility, and improved aspects of pancreatic cancer organoids, making them indispensable models to fill the gap between in vitro and in vivo models for precious medicine.
These viewpoints in this review are reasonably supported. However, the following major points need to be addressed.
Major points
1. In the section “Organoid”, contents describing the history of overall organoids are recommended to be compressed since this review should focus on pancreatic cancer organoids rather than overall organoids.
2. In the section “Methods of PDAC organoids development”, the last paragraph regarding “how we choose PDO or PDOX for our different purpose” is recommended to be added/discussed for experimental guidance.
3. In the section “Tumor organoids in precision medicine for PDAC”, more aspects of precious medicine like target screening and tumor signaling pathways, besides the drug screening through pharmacogenomics, are recommended to be added.
Author Response
We thank the reviewer for the effort she/he put to review our manuscript, the encouraging words and the constructive comments that helped us to improve and strengthen our manuscript. Please find below our responses to reviewer’s specific comments.
This review demonstrates an overview of pancreatic cancer organoids for precious medicine. It systematically and explicitly illuminated the necessity, feasibility, and improved aspects of pancreatic cancer organoids, making them indispensable models to fill the gap between in vitro and in vivo models for precious medicine.
These viewpoints in this review are reasonably supported. However, the following major points need to be addressed.
Major points
1. In the section “Organoid”, contents describing the history of overall organoids are recommended to be compressed since this review should focus on pancreatic cancer organoids rather than overall organoids.
Following the reviewer's suggestion we have now shortened this section.
2. In the section “Methods of PDAC organoids development”, the last paragraph regarding “how we choose PDO or PDOX for our different purpose” is recommended to be added/discussed for experimental guidance.
We apologize to the reviewer but we could not find a clear guideline as to how we choose between PDO or PDOX. However and since this is an important issue we have added the following in the discussion section (lines 611-616):
"Their use in the development of more clinically relevant animal models of cancer is another field where organoids gain ground. PDO xenografts are reported to better represent the cancer cell state of the parental tumor, and thus serve as more effective models for drug testing, they can be easily genetically modified, allowing for broad applications in basic cancer research and show a higher success rate as compared to PDX models at a more reasonable time and cost {for a more comprehensive review on this see [67]}.
3. In the section “Tumor organoids in precision medicine for PDAC”, more aspects of precious medicine like target screening and tumor signaling pathways, besides the drug screening through pharmacogenomics, are recommended to be added.
Following the reviewer's recommendations we have added the following
In section "5. Tumor organoids in precision medicine"
(lines 393-397) "Targeted agents like selumetinib (MEK inhibitor), afatinib, everolimus, and LY2874455 (fibroblast growth factor receptor inhibitor) were tested. Authors found that PDOs harboring ERBB2 amplifications and EGFR mutations were sensitive to the tyrosine kinase inhibitor afatinib, whereas a PDO carrying an oncogenic PIK3CA allele was sensitive to the mTOR agent everolimus."
(lines 410-414) "In the same study, authors used PDOs to evaluate the potential of a new targeted agent, EZP01556, as an anticancer agent. This inhibitor specifically targets the arginine methyltransferase 5 (PRMT5) protein. PRMT5 is reported as a synthetic lethal gene in methylthioadenosine phosphorylase deficient (MTAP−) cells [51]."
In section "6. Discussion" (lines 595-610)
"The investigation of mechanisms that result to drug resistance and strategies for overcoming them is a field of immense use of PDOs. Ponz-Sarvise et al., in an effort to study and overcome the resistance observed in PDAC upon combined MAPK and PI3K inhibition, identify the overactivation of ERBB as a critical parameter in this pathway. They further demonstrate that the combined MEK and ERBB inhibition using the MEK inhibitor selumetinib and the pan-ERBB inhibitor neratinib of human organoid orthotopic xenografts resulted to tumor regression in short-term intervention studies [64]. Another example is the investigation and the comprehension of the underlying mechanism of resistance development of PDAC against FOLFIRINOX regimen, as demonstrated in a recent study by Bachir et al., with the aid of PDOs [65]. Besides the use of PDOs for drug studies, these models are already in use to uncover and validate novel targets and signaling pathways involved in pancreatic cancer progression [47, 48, 51, 66]. Organoids are extensively used to shed light in the mechanisms driving the metastatic potential of PDAC. With the aid of PDOs, Huang et al., identified SMAD4 as the transcription factor responsible for promoting EMT and orchestrates the invasion program in PDAC [67]."
We hope that we have addressed properly all the concerns raised by the reviewer. Than you again
Round 2
Reviewer 3 Report
This revised review is sound and works a significant contribution to the field of pancreatic cancer organoids.